# Quantitative Metabolomics of Tissue, Perfusate, and Bile from Rat Livers Subjected to Normothermic Machine Perfusion

**DOI:** 10.3390/biomedicines10030538

**Published:** 2022-02-24

**Authors:** Caterina Lonati, Daniele Dondossola, Laimdota Zizmare, Michele Battistin, Leonie Wüst, Luigi Vivona, Margherita Carbonaro, Alberto Zanella, Stefano Gatti, Andrea Schlegel, Christoph Trautwein

**Affiliations:** 1Center for Preclinical Research, Fondazione IRCCS Ca’ Granda Ospedale Maggiore Policlinico, Via Pace 9, 20100 Milan, Italy; caterina.lonati@gmail.com (C.L.); michele.battistin@policlinico.mi.it (M.B.); stefano.gatti@policlinico.mi.it (S.G.); 2General and Liver Transplant Surgery Unit, Fondazione IRCCS Ca’ Granda Ospedale Maggiore Policlinico, Via Francesco Sforza 35, 20100 Milan, Italy; dondossola.daniele@gmail.com (D.D.); margherita.carbonaro@unimi.it (M.C.); 3Department of Pathophysiology and Transplantation, University of Milan, Via Francesco Sforza 35, 20100 Milan, Italy; luigi.vivona@unimi.it (L.V.); alberto.zanella1@unimi.it (A.Z.); 4Werner Siemens Imaging Center, Department of Preclinical Imaging and Radiopharmacy, University Hospital Tübingen, Eberhard Karls University of Tübingen, Röntgenweg 13, 72076 Tübingen, Germany; laimdota.zizmare@med.uni-tuebingen.de (L.Z.); leo-h-wuest@t-online.de (L.W.); 5Department of Anesthesia and Critical Care, Fondazione IRCCS Ca’ Granda Ospedale Maggiore Policlinico, Via Francesco Sforza 35, 20100 Milan, Italy; 6Department of Surgery and Transplantation, Swiss HPB Centre, University Hospital Zurich, 8091 Zürich, Switzerland; schlegel.andrea@outlook.de

**Keywords:** nuclear magnetic resonance (NMR) spectroscopy, liver machine perfusion (MP), ischemia/reperfusion (IR), mitochondrial metabolism, tricarboxylic acid (TCA) cycle, preclinical research, ketogenesis, succinate, 3-hydroxybutyrate, O-phosphocholine

## Abstract

Machine perfusion (MP) allows the maintenance of liver cells in a metabolically active state ex vivo and can potentially revert metabolic perturbations caused by donor warm ischemia, procurement, and static cold storage (SCS). The present preclinical research investigated the metabolic outcome of the MP procedure by analyzing rat liver tissue, bile, and perfusate samples by means of high-field (600 MHz) nuclear magnetic resonance (NMR) spectroscopy. An established rat model of normothermic MP (NMP) was used. Experiments were carried out with the addition of an oxygen carrier (OxC) to the perfusion fluid (OxC-NMP, *n* = 5) or without (h-NMP, *n* = 5). Bile and perfusate samples were collected throughout the procedure, while biopsies were only taken at the end of NMP. Two additional groups were: (1) Native, in which tissue or bile specimens were collected from rats in resting conditions; and (2) SCS, in which biopsies were taken from cold-stored livers. Generally, NMP groups showed a distinctive metabolomic signature in all the analyzed biological matrices. In particular, many of the differentially expressed metabolites were involved in mitochondrial biochemical pathways. Succinate, acetate, 3-hydroxybutyrate, creatine, and O-phosphocholine were deeply modulated in ex vivo perfused livers compared to both the Native and SCS groups. These novel results demonstrate a broad modulation of mitochondrial metabolism during NMP that exceeds energy production and redox balance maintenance.

## 1. Introduction

The introduction of machine perfusion (MP) significantly changed the clinical scenario of liver transplantation by enabling prolonged ex situ preservation, evaluation, and reconditioning of grafts previously deemed unsuitable [1,2]. In fact, relative to static cold storage (SCS), ex vivo perfusion allows a paradigm shift from functional suppression to metabolic activation [3,4]. This condition could promote the induction of beneficial endogenous pathways to cope with the detrimental consequences of ischemia/reperfusion (IR) [5].

Cellular and mitochondrial metabolism plays a crucial role in IR injury [6,7,8]. Indeed, cell survival directly depends on the degree of metabolic depression caused by ischemia [9]. Of note, activation of specific metabolic events within cells and mitochondria can contribute to a poor transplant outcome [8,10]. In particular, unbalanced accumulation of mitochondrial metabolites was indicated as a key event in cell deterioration, through the induction of reactive oxygen species (ROS) generation and subsequent cell death following reperfusion [6,7,8,11,12,13]. Metabolomic profiling of liver biopsies collected at different stages of liver transplantation provided useful information to determine the extent of organ injury related to cold ischemia [14], to assess the recovery rate post transplantation [14,15], and to identify prognostic markers of graft and dysfunction and acute rejection [16,17,18].

Based on this, the description of liver metabolic states during MP emerges as an essential prerequisite to fully take advantage of the reconditioning potential of this technique and, consequently, to improve its clinical application.

The present research aim was to determine the metabolic profiles of ex vivo perfused livers by means of high-field (600 MHz) nuclear magnetic resonance (NMR) spectroscopy-based metabolomics of tissue biopsies, perfusate, and bile samples. Since the large biological variation intrinsic in human donor samples could mask the metabolic changes induced by the perfusion procedure, we used an established model of rat liver normothermic MP (NMP) [19]. In addition, based on our previous investigation demonstrating that the amount of oxygen delivered during NMP deeply influences hepatocyte metabolism and bile production [19], we opted to perform NMP experiments either with the addition of an oxygen carrier (OxC) to the perfusion fluid or without it.

The present study performed a quantitative metabolomics investigation of tissue, perfusate, and bile from rat livers subjected to ex vivo perfusion. The results of this research could contribute to disclose whether and how NMP modulates the deleterious metabolic events induced by IR, while providing useful information concerning bile composition during NMP.

## 2. Materials and Methods

### 2.1. Animals and Study Design

The preclinical procedures were performed at the Center for Preclinical Research under the authorization number 60/2019, granted by the Italian Institute of Health (issuing date 28 January 2019). Animals received humane care in compliance with the Principles of Laboratory Animal Care and the experiments were conducted according to the Animal Research: Reporting of In Vivo Experiments (ARRIVE) guidelines [20].

Twenty adult Sprague-Dawley male rats weighing 240–330 g (Envigo RMS. S.R.L, Udine, Italy) were used. Animals were housed in a ventilated cage system (Tecniplast S.p.A., Varese, Italy) at 22 ± 1 °C, 55 ± 5% humidity, with a 12 h dark/light cycle and access to chow feed ad libitum.

A schematic workflow diagram of the investigation is shown in Appendix A. Rats were randomly assigned to one of the following experimental groups: (1) Native (*n* = 10), in which either liver biopsies (*n* = 5) or bile samples (*n* = 5) were collected immediately after laparotomy of rats in resting conditions; (2) SCS (*n* = 5), in which livers were subjected to in situ cold flushing with Celsior solution (Institut Georges Lopez, Lissieu, France), procured and subjected to 30 min of cold (4 °C) ischemia; (3) Hypoxic NMP (h-NMP) (*n* = 5), in which livers were subjected to the same procedures as the SCS group and then ex vivo perfused for 150 min; (4) OxC-NMP (*n* = 5), in which livers were subjected to the same procedures of the SCS group and then ex vivo perfused for 150 min using a perfusion fluid supplemented with an OxC.

### 2.2. Surgical Procedure

Rats were anesthetized with intraperitoneal injection of sodium thiopental (Inresa Arzneimittel, Freiburg, Germany) and maintained by spontaneous breathing in an oxygen mask until sacrifice, as previously described [21,22]. Next, the abdomen was opened and the bile duct cannulated. Systemic heparinization (500 UI) was then performed, while the portal vein was cannulated with a 16G cannula and connected to the in situ perfusion circuit. A sternotomy was performed to remove the heart and to cut the inferior vena cava. Next, the liver was flushed in situ with 35 ml of Celsior solution at 4 °C with a mean pressure of 22 mmHg. After procurement, the liver was cold-stored in 4 °C Celsior solution for 30 min. Thereafter, the portal vein cannula was connected to the perfusion circuit and the NMP was started.

### 2.3. Normothermic Machine Perfusion (NMP) Protocol

The NMP perfusion system (Appendix A) adopted in this study was previously described in detail [19].

The perfusion fluid was prepared with Dulbecco’s Modified Eagle Medium (DMEM, Thermo Fisher Scientific, Waltham, MA, USA), 20% albumin (Immuno Baxter S.p.A., Roma, Italy), antibiotics and antimycotics (Thermo Fisher Scientific), and insulin (see Appendix A for details). Experiments with the OxC-NMP group involved the addition of an OxC to the perfusion fluid. In particular, we used Oxyglobin (HbO_2_ Therapeutics, Boston, MA, USA), a veterinary-licensed non-cellular bovine-derived haemoglobin.

The NMP-protocol lasted 150 min and included two phases: (1) a “rewarming phase”, representing the first 30 min of liver perfusion. In this phase, perfusion fluid temperature and portal flow were gradually increased until their target values were reached; (2) a “normothermic phase”, during which temperature was maintained at 37 °C and portal flow was kept at 30 mL/min to obtain a maximal portal pressure of 8 mmHg.

### 2.4. Collection, Processing, and Analysis of Perfusate and Bile Samples

Perfusate was sampled from the portal vein and inferior vena cava at 30 min, 90 min, and 150 min. An aliquot of the perfusate samples was immediately used to assess acid–base balance, electrolytes, and metabolites using a gas analyzer (ABL 800 Flex, A. De Mori Strumenti, Milan, Italy). Oxygen delivery (DO_2_) and oxygen consumption (VO_2_) were calculated as previously described [19]. After 10 min centrifugation at 600× *g* at 4 °C, aliquots of perfusate were stored at −80 °C for subsequent assessment of alanine aminotransferase (ALT) and lactate dehydrogenase (LDH).

Bile was collected in a 1.5 mL tube placed 10 cm below the liver graft and it was sampled every hour after rewarming. Vials were weighted with an analytical balance (KERN & SOHN GmbH, Balingen, Germany) to assess the amount of bile produced over the NMP procedure. Additionally, bile was collected in vivo from rats in resting conditions. Briefly, rats (*n* = 5) were anesthetized and then subjected to a laparotomy and bile duct cannulation to recover 1.5 mL of bile fluids from each rat. Bile samples were stored at −80 °C for subsequent analysis.

### 2.5. Tissue Sample Collection and Analyses

Snap-frozen liver biopsies were used to evaluate the adenosine triphosphate (ATP) content. At the end of the procedure, the median lobe was weighed with an analytical balance (KERN & SOHN GmbH) and dried at 50 °C for 24 h (LTE Scientific, Greenfield, UK). The wet-to-dry ratio (W/D) was calculated and used as an index of edema.

### 2.6. Nuclear Magnetic Resonance (NMR) Spectroscopy-Based Metabolomics

NMR spectroscopy-based metabolomic analysis was performed in liver tissue homogenates, perfusate, and bile samples.

Deep-frozen liver tissue samples were cryogenically pulverized (Covaris cryoPREP CP02, Woburn, MA, USA) and subjected to ultra-sonication-based metabolite extraction in a liquid state (Covaris E220 Evolution, Woburn, MA, USA). Liver powder was transferred to a Covaris system-compatible glass tube, suspended in 300 μL LC–MS purity grade methanol (Sigma-Aldrich Chemie, Taufkirchen, Germany) and 1000 μL *tert*-butylmethyl ether (Sigma-Aldrich Chemie, Taufkirchen, Germany) and subjected to the metabolite extraction procedure. After ultra-sonication, 250 μL of molecular biology purity-grade water was added and tubes were centrifuged for phase separation (12,000× *g* for 20 min). The aqueous layer was separated, transferred to a 1.5 mL Eppendorf cup (Eppendorf SE, Hamburg, Germany), and evaporated to dryness overnight. An aliquot of 200 μL from each perfusate and bile sample was also evaporated to dryness overnight.

The dried metabolite pellets were re-suspended in routine deuterated phosphate buffer solution (pH = 7.4, 1M K_2_HPO_4_ (Sigma-Aldrich Chemie, Taufkirchen, Germany), 10 mM NaN_3_ (Sigma-Aldrich Chemie, Taufkirchen, Germany)) containing 1 mM internal NMR spectroscopy reference standard 3-(trimethylsilyl) propionic-2,2,3,3-d_4_ acid sodium salt (TSP) (Sigma-Aldrich Chemie, Taufkirchen, Germany) for quantification. NNR suspensions were centrifuged (30,000× *g* for 30 min) to remove any undissolved particles and poured into 3 mm Bruker SampleJet-compatible NMR spectroscopy sample tubes (Bruker BioSpin, Ettlingen, Germany).

After spectra acquisition for 1 h for bile and perfusate samples and between 1 and 4 h for liver tissue (depending on the amount of liver powder) with Carr–Purcell–Meiboom–Gill (CPMG) pulse sequence using a Bruker Avance III HD 14.10 Tesla spectrometer (600 MHz for ^1^H) with a 1.7 mm triple-resonance room-temperature probe (Bruker BioSpin, Ettlingen, Germany), spectra were pre-processed with Bruker TopSpin 3.6.1 software, quantified using ChenomX NMR suite 8.5 software, and exported as a concentration table.

### 2.7. Statistical Analysis

Data are presented as means ± SEM or median (25–75th percentiles). Differences across experimental groups were investigated using the one-way analysis of variance (ANOVA) or two-way repetitive measures analysis of variance (two-way RM ANOVA). Tukey’s post hoc test was used for multi-comparison procedures. Non-normally distributed data were rank-transformed before the performance of ANOVA. Uptake ratios of perfusate parameters were calculated as ((Cstart−Cend)/Cstart). A probability value of *p* < 0.05 was considered statistically significant. All the tests were performed using SigmaStat software 3.5 (Systat Software Inc, San Jose, CA, USA).

With regard to NMR data, statistics were analyzed using the MetaboAnalyst 5.0 tool package online platform ([23] www.metaboanalyst.ca, accessed on 28 January 2022) and Prism Graph Pad 9.1.0 software (San Diego, CA, USA). Briefly, we normalized the dataset with a randomly selected reference sample using the probabilistic quotient normalization (PQN) approach, accounting for dilution. Partial least-squares discriminant analysis (PLS-DA) maximizes the covariance and shows the first two components on an X- and Y-axis with a 95% confidence region. Then, a parametric analysis of variance (ANOVA) with an adjusted *p*-value (FDR) cut-off of 0.05 with Fisher’s LSD post hoc analysis was performed. For the heat maps, we used a Euclidean distance measure with a Ward clustering algorithm. For bile and perfusate intermatrix comparisons, the PQN normalization was applied using a pooled sample group (OxC-NMP end group) and further log 10 transformation to normalize the data variation between the two matrices. Venn diagrams were created using the Bioinformatics and Evolutionary Genomics software at https://bioinformatics.psb.ugent.be/webtools/Venn (accessed on date 7 January 2022). 

## 3. Results

### 3.1. Baseline Perfusate Composition

Baseline perfusate composition was different in the two NMP groups due to the presence of different compounds necessary to preserve oxyglobin (Appendix A).

### 3.2. Oxygen Delivery/Consumption during NMP

DO_2_ was lower in the NMP group compared to the OxC-NMP group (*p* < 0.001) (Appendix A). VO_2_ was higher in the livers perfused with an OxC over the first 60 min of normothermic phase, while it became similar across the experimental groups (*p* = 0.032) in later stages. DO_2_ and VO_2_ showed a positive correlation (r^2^ = 0.820, *p* < 0.001).

### 3.3. Liver Graft Viability and Function

Perfusate ALT concentration increased throughout the NMP procedure in both the NMP and the OxC-NMP groups (NMP: from 1.07 ± 0.69 U/L/g liver to 2.66 ± 1.21 U/L/g liver, *p* = 0.029; OxC-NMP: from 0.87 ± 0.71 U/L/g liver to 2.27 ± 1.27 U/L/g liver, *p* = 0.027). The potassium uptake ratio was higher in the perfusate from the OxC-NMP group compared to that observed in the NMP group (NMP: 0.04 ± 0.05 vs. OxC-NMP: 0.12 ± 0.02, *p* = 0.002). Furthermore, lactate concentration in the OxC-NMP group was lower than in the h-NMP perfusate at all times point of the normothermic phase (*p* = 0.016) (Figure 1A). Consistently, the lactate uptake ratios of livers perfused without the addition of an OxC were lower than those observed for the OxC-NMP group (NMP: 0.062 ± 0.359 vs. OxC-NMP: 0.819 ± 0.133, *p* = 0.016).

The amount of bile produced at 90 min and at 150 min was similar between the experimental groups (h-NMP 90 min: 0.028 mL, OxC-NMP 90 min 0.036 mL; h-NMP 150 min 0.053 mL, OxC-NMP 150 min 0.050 mL, *p* = 0.762).

Compared to the Native group, the ATP content of livers subjected to NMP was significantly lower (Figure 1B). However, while livers perfused without OxC showed the smallest amount of tissue ATP across all the experimental groups (*p* = 0.007), those from the OxC-NMP group had similar ATP concentrations to those observed in the SCS group (*p* = 0.410) (Figure 1B).

### 3.4. Unsupervised Analysis of Tissue, Perfusate, and Bile Samples

The overall effects exerted by NMP on the liver metabolomic phenotype were investigated by partial least-squares discriminant analysis (PLS-DA). To avoid introduction of any potential bias in the interpretation of results, all the metabolites which were detected in our dataset but also included in the Celsior solution (mannitol, histidine, glutamate, glutathione) and the DMEM medium (glycine, arginine, cysteine, glutamine, histidine, isoleucine, leucine, lysine, methionine, phenylalanine, serine, threonine, tryptophan, tyrosine, valine, choline, pantothenate, nicotinamide, inositol, glucose) were excluded from the study. Unsupervised analysis indicated broad changes in the metabolite concentrations of the NMP groups compared to the control groups in all the evaluated biological specimens.

Liver biopsies showed distinctive metabolite profiles across Native, SCS, h-NMP, and OxC-NMP groups. More specifically, there were substantial differences between the grafts subjected to ex vivo perfusion and non-perfused livers (Figure 2A, “Tissue”). Among NMP groups, each specific experimental condition could be identified based on metabolite profiles, with no overlaps between them in the sPLS-DA scores plot. We annotated and quantified 56 metabolites involved in different metabolic pathways, most of which take place inside mitochondria: tricarboxylic acid (TCA) cycle and oxidative phosphorylation (OXPHOS), choline metabolism, creatine turnover, phospholipid metabolism, methionine (Met) cycle, one-carbon metabolism, cysteine (Cys) oxidation, NAD^+^ turnover, ß-oxidation, ketogenesis, glycolysis, pyrimidine metabolism, and UDP-sugars synthesis (Appendix A). The perfusate analysis clearly discriminates between the h-NMP and the OxC-NMP groups and discloses that the metabolite concentrations were modulated throughout NMP (Figure 2A, “Perfusate” and Appendix A). With regard to bile, unsupervised analysis showed a marked separation between the samples collected during NMP and the native bile (Figure 2B). In addition, bile composition varied over time (Appendix A).

The metabolites showing the highest differences across experimental groups are listed in the sPLS-DA loadings plots in Figure 2B. Venn diagrams shows how the top 20 differentially expressed metabolites overlap between each biological sample analyzed (Figure 2C). This analysis indicated that acetate, succinate, creatine, and O-phosphocholine were markedly modulated in liver tissue, perfusate, and bile samples.

### 3.5. Metabolite Changes in Liver Biopsies and Perfusate Samples

The metabolites with the highest variation within groups in both tissue and perfusate samples were creatine, succinate, acetate, O-phosphocholine, alanine (Ala), methylguanidine (MG), 1-methylnicotinamide (1-MNA), and acetoin (Figure 2B).

The heatmaps reported in Appendix A show all the metabolites differentially expressed in either liver homogenates or perfusates. Among these, lactate tissue content was significantly increased in the h-NMP and OxC-NMP groups compared to non-perfused livers (Native: 1.969 ± 0.103; SCS: 2.604 ± 2.096; h-NMP: 4.203 ± 0.856; OxC-NMP: 3.887 ± 1.373; *p* = 0.004). Consistently, the corresponding amino acid Ala showed a similar trend across experimental groups (Native: 0.352 ± 0.803; SCS: 0.781 ± 0.333; h-NMP: 1.578 ± 0.342; OxC-NMP: 1.363 ± 1.221, *p* = 0.007). GSSG liver content was similar between the Native and OxC-NMP groups, while it was reduced in livers subjected to NMP without OxC (Native: 0.332 ± 0.076; SCS: 0.212 ± 0.046; h-NMP: 0.266 ± 0.071; OxC-NMP: 0.356 ± 0.110, *p* = 0.002).

Concerning mitochondrial metabolites, they can be clustered in six biochemical pathways (Figure 3): (A) TCA cycle and OXPHOS; (B) fatty acid (FA) metabolism and ketogenesis; (C) creatine metabolism; (D) NAD^+^ turnover; (E) choline metabolism; (F) pyrimidine metabolism and UDP sugars. Succinate was higher in h-NMP liver homogenates compared to native livers, while similar concentrations were observed in the OxC-NMP and non-perfused groups (Figure 3A). Consistently, the perfusate succinate concentration was lower in the OxC-NMP group compared to the h-NMP group (Figure 3A). Tissue acetate content was greater in the grafts subjected to NMP compared to both control groups (*p* < 0.05) (Figure 3B). Of note, a further increase in this metabolite was observed in the OxC-NMP group compared to the grafts perfused without the addition of an OxC. Concentration of 3-hydroxybutyrate (3-HB) was considerably enhanced in grafts subjected to NMP, with no differences attributable to the presence of the OxC (Figure 3B). Liver creatine was reduced in all the grafts subjected to NMP compared to non-perfused livers (Figure 3C). However, the OxC-NMP group was associated with higher perfusate concentrations of this metabolite compared to the h-NMP group at all time points. The tissue content of nicotinamide ribotide (nicotinamide mononucleotide, NMN), a precursor of NAD^+^ biosynthesis, was significantly higher in livers subjected to NMP with OxC compared to all the other experimental groups (Figure 3D). Perfusate concentrations of the metabolite 1-MNA were consistently reduced in the OxC-NMP group compared to those observed in perfusates drained from livers from the h-NMP group (Figure 3D). O-phosphocholine was significantly reduced in the OxC-NMP group compared to all the other experimental groups (Figure 3E). Sarcosine and dimethylamine concentrations were higher in the perfusates from the OxC-NMP group compared to those from the h-NMP group (Figure 3E). Tissue uracil was downregulated in ex vivo perfused livers (Figure 3F), while perfusate cytidine of the OxC-NMP group was lower compared to that of the h-NMP group (Figure 3F). Consistently, a decreased content of uridine diphosphate glucose (UDP-Glc) was observed in livers subjected to NMP compared to both control groups (Figure 3F). Tissue homogenates from the OxC-NMP showed the lowest content of UDP-glucuronate (UDP-glucuronic acid, UDP-GA) (Figure 3F).

### 3.6. Metabolite Changes in Bile Samples

To understand how bile metabolite concentrations changed throughout the NMP procedure, an additional PLS-DA analysis of bile metabolomic data was performed, excluding native bile samples (Figure 4A). Hereby, fourteen molecules were found to be modulated when compared with in vivo bile (Figure 2) and also with the h-NMP and OxC-NMP groups: 3-hydroxyphenylacetate (3-HPA), C-25 taurine conjugate, lactate, NMN, GSSG, betaine, N,N-dimethylglycine (DMG), formate, O-phosphocholine, succinate, glycerol, acetate, GA, and the N(CH_3_)_3_ phospholipid choline head group of bile acids. The heatmap shows the bile metabolites that were significantly changed across all the experimental groups and within the NMP groups over time (Figure 4B). 

Table 1 illustrates how the concentration of these metabolites changed over the NMP procedure compared to in vivo bile. In particular, five classes of modulation were identified: (1) increased compared to bile collected in vivo; (2) decreased compared to bile collected in vivo; (3) increased at the “start” and restored at the “end” of the NMP procedure; (4) unchanged compared to bile collected in vivo; (5) differentially modulated in the h-NMP and OxC-NMP groups. Of note, acetate, glycerol, and 3-HPA showed a different profile compared to the native bile at the beginning of NMP, but were restored to in vivo levels at the end of the procedure.

### 3.7. Comparison between Metabolite Concentration in Bile and Perfusate Samples

The effects of NMP on liver metabolism were further investigated by comparing metabolite concentrations at the beginning and at the end of the procedure in both bile and perfusate samples (Figure 5A). Ten of the metabolites with the highest differences across groups in bile samples were found to be significantly changed, also, in the perfusate samples: fumarate, acetate, carnitine, aspartate (Asp), formate, acetoin, succinate, DMG, O-phosphocholine, and ala (Figure 5A). 

Figure 5B shows the metabolites affected by NMP in bile and perfusate: lactate and acetate concentration markedly decreased over NMP performed using an OxC ([C_End_]/[C_Start_] log10 < 1), while a slighter reduction was observed in the h-NMP group. An opposite trend was shown for 3-HB and O-phosphocholine (except for bile samples using an OxC, where a decrease was observed for 3-HB), with greater increases in both bile and perfusate in the h-NMP group compared to the OxC-NMP group ([C_End_]/[C_Start_] log10 > 1).

## 4. Discussion

The present research performed, for the first time, a comprehensive metabolomics evaluation of rat liver grafts subjected to NMP by analyzing tissue, bile, and perfusate samples by high-field ^1^H-NMR spectroscopy and subsequent metabolomic analysis. The metabolic signature of tissue biopsies and bile samples derived from ex vivo perfused livers showed substantial differences compared to both the in vivo condition and cold ischemia alone. In particular, there was a marked modulation of several metabolites involved in different biochemical pathways taking place within mitochondria.

Clinical and preclinical studies suggest that the positive effects exerted by MP on liver grafts could depend on the ability of the procedure to promote a reversal of the metabolic perturbations induced by ischemia during organ retrieval and storage [8,10,11,17,24,25]. Therefore, to gain insight into the beneficial properties of NMP, we explored the metabolic phenotype of ex vivo perfused livers by profiling liver tissue, bile, and perfusate samples. To capture the metabolic influences of NMP per se, experiments were carried out using uninjured healthy livers, in order to avoid the introduction of any disease-related confounders.

Our analysis shows that NMP is indeed associated with broad changes in the metabolite profiles of liver grafts compared to the native liver pattern. This modulatory action appears to be specific to the perfusion procedure, which involves different metabolites to those altered by cold ischemia alone. Of note, the metabolic effect exerted by NMP on liver parenchyma influenced, in a time-dependent manner, both perfusate and bile composition.

The main finding is that most of the differentially expressed metabolites are intermediates of biochemical reactions catalyzed by enzymes located within mitochondria. As presented in Figure 6, significant changes were observed in the following mitochondrial pathways: TCA cycle, OXPHOS, ß-oxidation, ketogenesis, methylamine metabolism, pyrimidine metabolism, and creatine metabolism. Among the modulated metabolites, succinate, acetate, O-phosphocholine, and creatine showed substantial changes in their concentrations in all the biological samples evaluated.

In line with previous studies [15,25,26], we found specific alterations in the TCA cycle and the electron transport chain in perfused livers. In particular, the rise of succinate observed in the NMP groups compared to native livers strongly suggests an incomplete recovery of OXPHOS during early NMP up to 150 min. The accumulation of this TCA intermediate was previously indicated as a key event leading to reverse electron transport and subsequent ROS production during post-ischemic reperfusion [6,7,27]. Of note, along with succinate increase, a reduced content of fumarate and oxaloacetate was observed in livers subjected to NMP relative to the Native group. The selective rise of succinate has already been described in livers subjected to warm ex vivo perfusion [25] and could be caused by a deranged activation of multiple metabolic reactions that converge on complex II succinate dehydrogenase (SDH) [27,28].

The increased acetate content found in perfused livers is likely derived from the utilization of fatty acids as an energy source to cope with ischemia-induced ATP depletion. Massive release of acetate could promote the induction of ketogenesis and subsequent 3-HB biosynthesis. In addition to being an essential energy substrate under stress conditions, 3-HB can exert multiple positive effects in cells exposed to ischemia, including sirtuin activation, resolution of inflammation, maintenance of cellular redox homeostasis, and induction of autophagy [29,30,31]. Therefore, 3-HB production during NMP could be considered as a beneficial event promoting cell recovery from ischemic damage.

Another remarkable observation of the present study is the substantial reduction of O-phosphocholine in ex vivo perfused compared to non-perfused livers. This phenomenon could signal a positive impact on later liver function because a higher O-phosphocholine content in tissue biopsies collected during cold storage was shown to be associated with early graft dysfunction (EAD) after transplantation [17,18]. The downregulation of O-phosphocholine during NMP could be a consequence of the modulation of the methylamine pathway and pyrimidine metabolism. In fact, utilization of phosphocholine in phospholipid biosynthesis depends on the availability of cytidine 5′-triphosphate (CTP) [32]. An altered pyrimidine biosynthesis in ex vivo perfused grafts is proved by the reduced content of uracil and UDP-conjugated sugars compared to native livers. Of note, although biosynthesis of pyrimidine mostly takes place in the cytosol, this pathway is linked with OXPHOS through the dihydroorotate dehydrogenase, a flavin mononucleotide (FMN)-containing enzyme located in the inner mitochondrial membrane [32]. Moreover, ischemia-dependent membrane lysis could account for the modulation of O-phosphocholine. Indeed, cell membrane disruption takes place in ischemic livers as a result of ATP depletion and activation of calcium-dependent phospholipases [14,15,17,26,33]. Consistently, different metabolites involved in the phospholipid turnover were found to be modulated in perfused compared to non-perfused grafts.

A reduced concentration of creatine was likewise observed in grafts subjected to NMP compared to both cold-stored and native livers. Under physiological conditions, creatine synthesis is achieved through two enzymatic reactions which occur mainly in the kidney and the liver, respectively [34]. Therefore, the lower creatine content found in ex vivo perfused livers could depend on the reduced efficiency of the first step required for creatine production due to the absence of kidney cell activity. Based on the importance of the creatine–phosphocreatine system in maintaining ATP levels during reperfusion of ischemic cells [35], this finding could indicate the need to improve perfusion fluid composition to maintain physiological creatine levels.

The bile produced by ex vivo perfused livers was dramatically different to native bile. In fact, bile-specific C-25 taurine conjugate and the phospholipid choline head group of bile acids were undetectable in samples collected during NMP. With regard to the other identified metabolites, we found both up- and downregulation compared to native bile, suggesting that NMP elicited specific biological processes in cholangiocytes, leading to selective excretion or reabsorption of intermediates. Of note, many of the differences marking the metabolic profile of the bile derived from the isolated liver were similar to those found in liver tissue homogenates. This further demonstrates that the metabolic state of parenchymal cells influences bile composition, which is in line with previous analyses of bile samples collected before and after liver transplantation [36]. In addition, both preservation (storage) and perfusion fluids can have a significant impact on bile production due to the different supply of nutritional factors compared to whole blood. Moreover, hepatic clearance of the compounds included in these solutions (e.g., mannitol, histidine) involves biliary excretion of specific metabolites [37]. Therefore, while changes in the amount of bile released during NMP are widely accepted as markers of viability [38,39], our results suggest that evaluation of bile composition in isolated livers needs to be further refined in that comparison with native bile does not appear entirely adequate.

Finally, significant differences were observed between the metabolic profiles of livers subjected to OxC-based NMP and those of the grafts perfused in the absence of OxC. The most interesting results include improved lactate clearance in perfusate and bile samples, higher ATP and acetate tissue content, lower succinate tissue and perfusate concentration, and reduced O-phosphocholine release in the OxC-NMP group. These data clearly indicate that the use of OxC leads to an improved energy status and enhanced functional recovery of liver grafts. In fact, lactate accumulation upon reperfusion can significantly affect graft viability and functional recovery [9,14,40,41,42,43,44], while a higher lactate concentration in blood samples of recipients was found to be associated with liver dysfunction [45]. In addition, the released lactate acts as damage-associated molecular patterns (DAMPs), mediating the inflammatory response at reperfusion [46]. With regard to liver energy homeostasis, the higher ATP content in the OxC-NMP group suggests a more efficient glycolysis and a faster recovery of OXPHOS [8,9,47]. Finally, lower succinate concentrations are not only indicative of a less severe impairment of the TCA cycle and OXPHOS in livers subjected to OxC-based NMP [28] but also lead to reduced tissue damage due to the crucial role of this metabolite in the induction of ROS production at complex I and in inflammation [6,8,48].

Some limitations can be mentioned for this study. The absence of transplantation procedures following NMP can be acknowledged as a weakness in the comparison between OxC-NMP and h-NMP groups. Indeed, the metabolic profiling of transplanted ex vivo perfused livers would have added significant information to better identify the differences marking OxC-based NMP and to understand whether the metabolic alterations induced by IR were fully restored by oxygenated NMP. Concerning NMR spectroscopy, the high salt contents in bile and perfusate samples resulted in line broadening, especially of the internal TSP standard, which requested log transformation to statistically investigate the metabolite data for these samples. Furthermore ^1^H-NMR only works for a selective portfolio of metabolites and applying mass spectrometry-based metabolomics would have yielded different metabolites, so that, whenever available, both NMR and MS should be used in future studies.

## 5. Conclusions

The results of the present study collectively indicate a broad modulation of mitochondrial metabolism during NMP. While some of the observed alterations are consistent with the current understanding of the metabolic perturbations induced by IR, our data reveal, for the first time, that mitochondrial anabolic and catabolic pathways are likewise modulated. These biochemical changes can influence several critical processes in liver cells, such as glucuronidation, glycogen synthesis, and bile production. Moreover, succinate, 3-HB, and other differentially expressed metabolites participate in leukocyte recruitment and activation, autophagy induction, and cell adaptation to oxidative stress. Therefore, the role of mitochondria in graft recovery following liver IR exceeds energy production and redox balance maintenance. This information can be applied as a useful starting point for future studies to discover biomarkers of liver viability and functional recovery in models reproducing the very many conditions of liver damage encountered in transplantation settings.

With regard to bile analysis, our study discloses a peculiar metabolic signature in samples produced by isolated livers. Targeted investigations are required to define “normal ranges” of metabolite concentrations in the bile released during NMP.

## Figures and Tables

**Figure 1 biomedicines-10-00538-f001:**
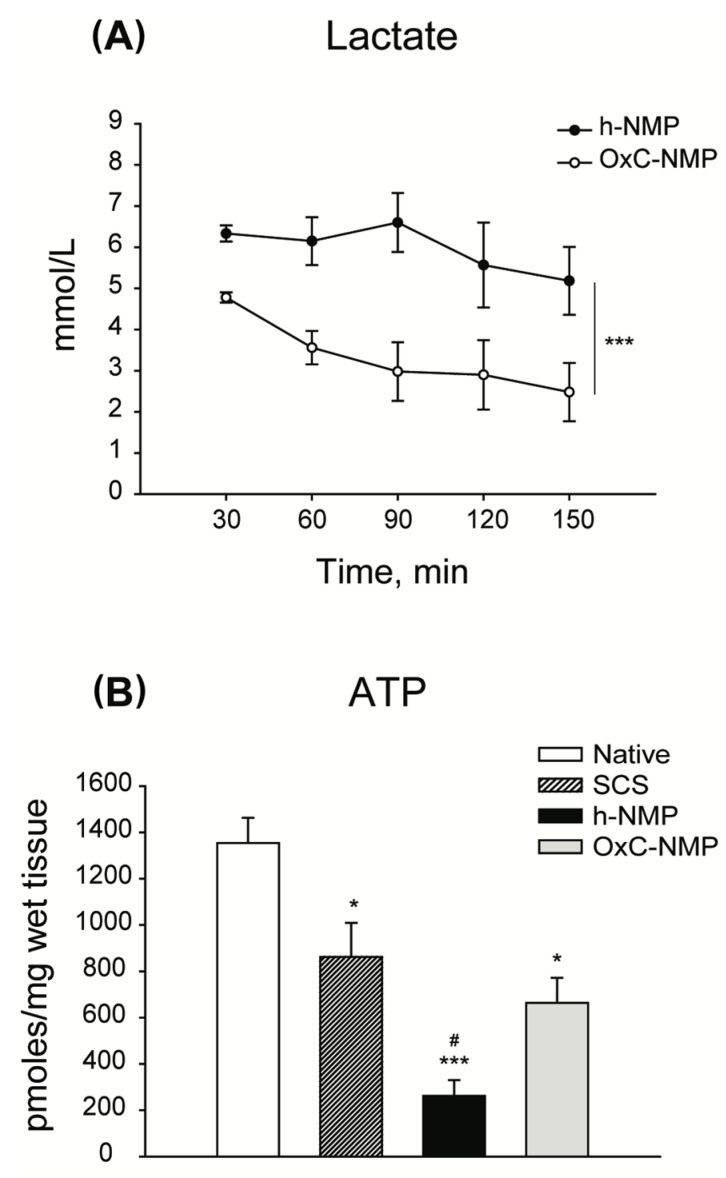
**Characterization of viability and function of livers subjected to normothermic machine perfusion (NMP).** (**A**) Perfusate lactate concentration throughout NMP. Two-way repeated measures ANOVA, *** *p* < 0.001. (**B**) Liver ATP content. ATP was measured in liver biopsies by means of bioluminescent assays. One-way ANOVA, Tukey’s post hoc test; *p* values vs. Native: * *p* < 0.05, *** *p* < 0.001; *p* values vs. SCS: # *p* < 0.05.

**Figure 2 biomedicines-10-00538-f002:**
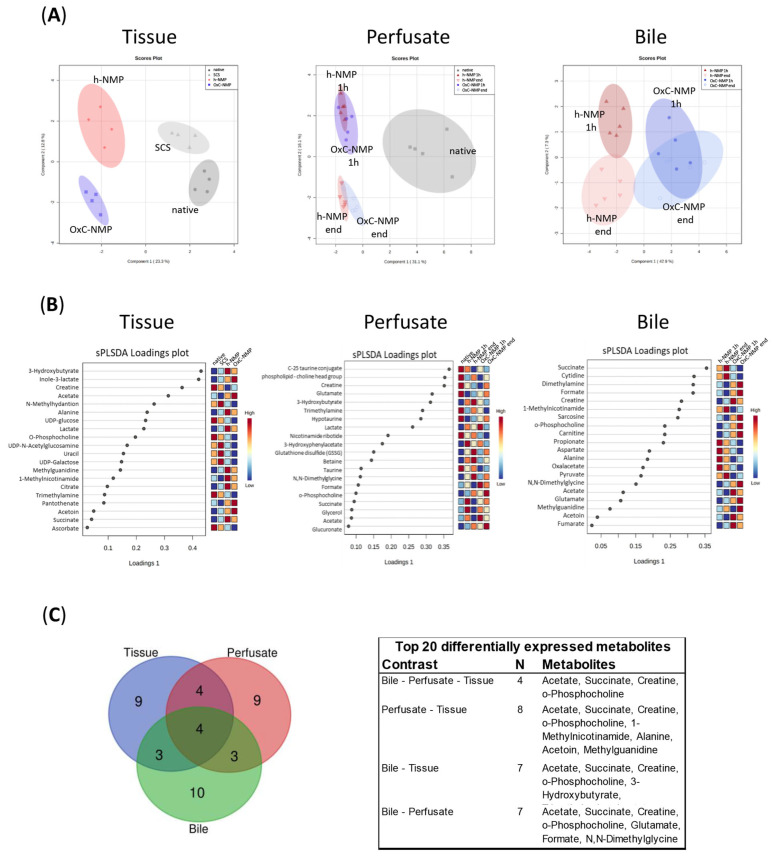
**Normothermic machine perfusion (NMP) metabolic profiling: unsupervised analysis of tissue, perfusate, and bile samples**. (**A**) sPLS-DA score plots of liver tissue, bile, and perfusate group separation; liver: Native—dark grey dots, SCS—light grey triangles, h-NMP—red rhombus, OxC-NMP—blue triangle. Bile and perfusate: Native—dark grey squares, h-NMP 1h—full red upwards triangles, h-NMP end—empty red downwards triangles, OxC-NMP 1h—full blue dots, OxC-NMP end—empty light blue circles. (**B**) sPLS-DA loadings plot with the top most significant metabolite changes observed in the liver tissue, bile, and perfusate analyses. (**C**) Venn diagram and corresponding table showing the top 20 differentially expressed metabolites in each of the analyzed biological samples and their distribution across tissue, perfusate, and bile specimens.

**Figure 3 biomedicines-10-00538-f003:**
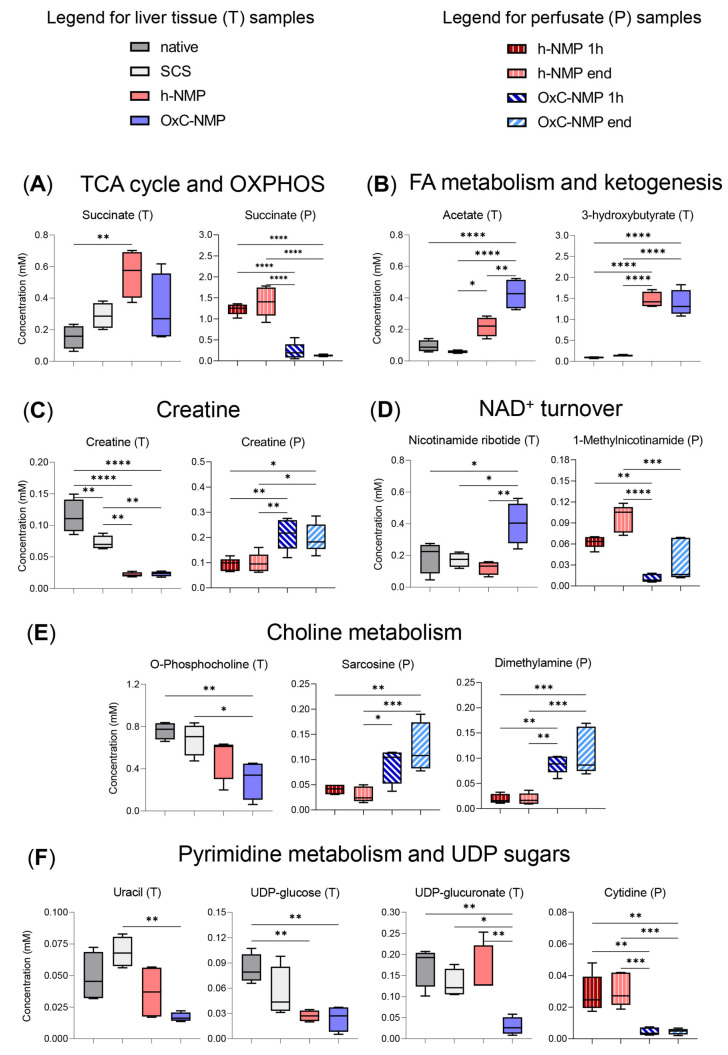
**Differentially expressed metabolites in liver tissue and perfusate samples.** Metabolic intermediates are grouped according to the following mitochondrial pathways: (**A**) tricarboxylic acid (TCA) cycle and oxidative phosphorylation (OXPHOS), illustrated by liver and perfusate succinate concentration box and whisker plot; (**B**) fatty acid (FA) metabolism and ketogenesis, illustrated by liver acetate and 3-HB; (**C**) creatine metabolism, illustrated by creatine concentration changes in liver tissue and perfusate; (**D**) NAD^+^ turnover, illustrated by liver nicotinamide ribotide and perfusate 1-methylnicotinamide (1-MNA) concentration changes; (**E**) choline metabolism, illustrated by liver O-phosphocholine and perfusate sarcosine and dimethylamine concentration changes; (**F)** pyrimidine metabolism and UDP sugars, illustrated by liver uracil, UDP-glucose, UDP-glucuronate, and perfusate cytidine concentration changes. Liver (*n* = 4) and perfusate (*n* = 5) box and whisker plots, one-way ANOVA with Tukey’s multiple comparison test, *p*-values: **** *p* < 0.0001, *** *p* < 0.001, ** *p* < 0.01, * *p* < 0.05. Liver data colors: Naive, dark grey; SCS, light grey; h-NMP light watercolor pink; OxC-NMP, light watercolor blue. Perfusate colors: h-NMP 1h, dark red, vertical striped pattern; h-NMP end, pink, vertical striped pattern; OxC-NMP 1h, dark blue, diagonal stripe pattern; OxC-NMP end, light bright blue, diagonal stripe pattern.

**Figure 4 biomedicines-10-00538-f004:**
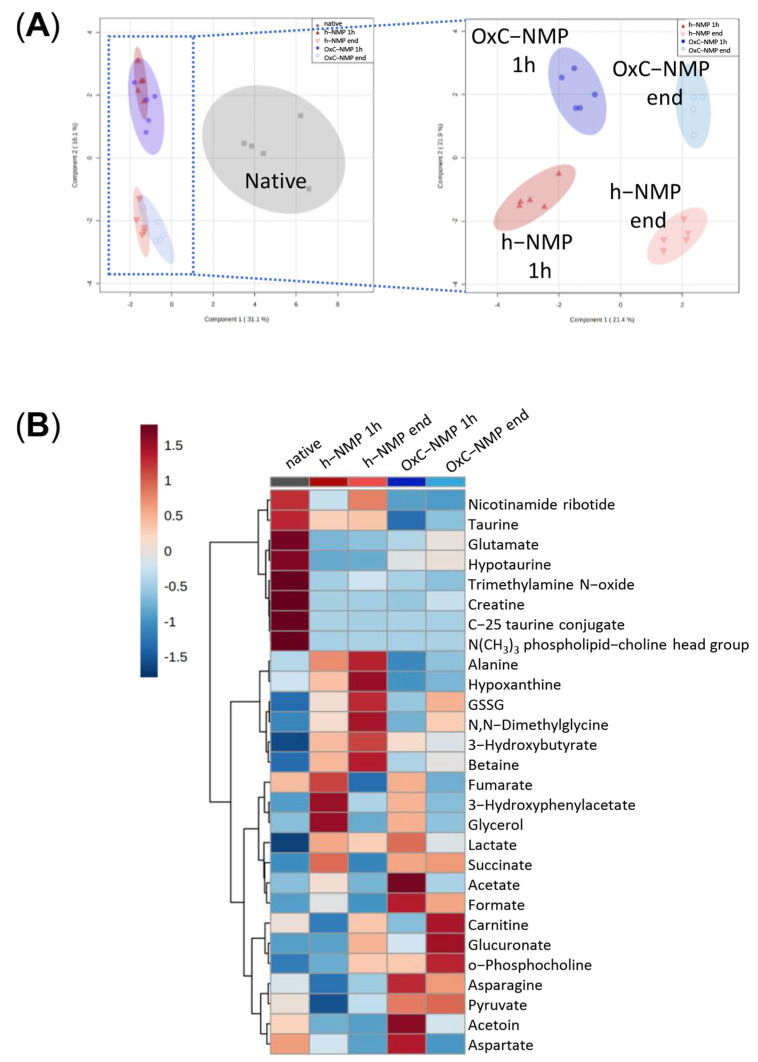
**Metabolite concentration changes in bile samples.** (**A**) sPLS-DA scores plots illustrating group separation based on sparse partial least-squares discriminant analysis. When excluding the Native group (dark grey squares), further separation of groups is achieved due to changes in metabolite concentrations in time; h−NMP 1h—dark red triangles, h−NMP end—light red downwards triangles, OxC−NMP 1h—dark blue full dots, OxC−NMP end—light blue empty dots. (**B**) Averaged heat map illustrating all the group metabolite changes—overview average for each group. Red—relatively increased concentration, blue—relatively reduced concentration. Abbreviation: GSSG, glutathione disulfide (oxidized form).

**Figure 5 biomedicines-10-00538-f005:**
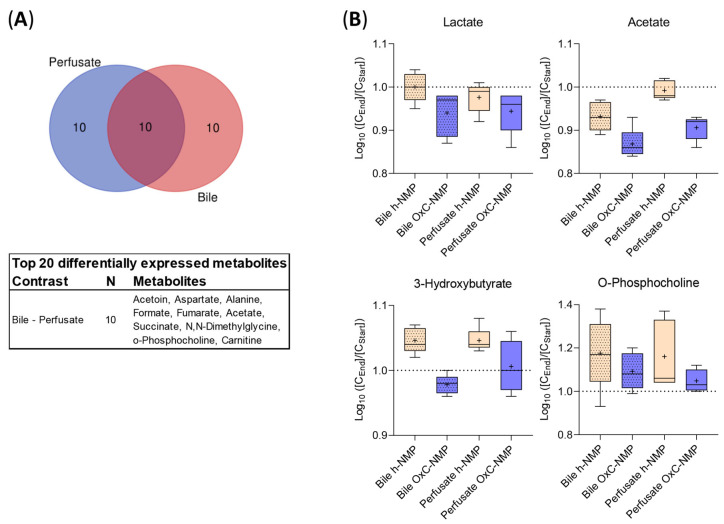
**Comparison between metabolite concentrations in bile and perfusate samples.** (**A**) Venn diagram and corresponding table showing the top 20 differentially expressed metabolites in either perfusate or bile samples and their distribution across the analyzed samples. (**B**) Normalized concentration ratio [C_End_]/[C_Start_] to logarithmic scale (log10) of metabolite increase (above 1.0 dashed line) or decrease (below 1.0 dashed line) during the perfusion in bile (dotted) and perfusate (clear) at 1 h (yellow) and at the end of NMP (blue), illustrated as bar plots with Tukey’s whiskers; line indicates the median, plus indicates the mean.

**Figure 6 biomedicines-10-00538-f006:**
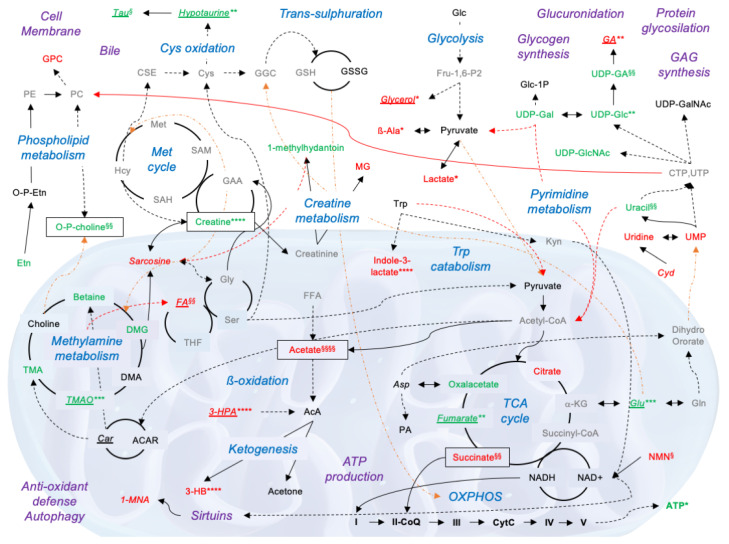
**Simplified diagram showing the metabolomic pathways modulated during normothermic machine perfusion (NMP) and their interactions.** Metabolites typed in red fonts are upregulated in the NMP groups compared to the Native group, while those typed in green fonts are downregulated. Metabolites typed in black fonts had a similar concentration across ex vivo perfused and native livers. Metabolites detected only in perfusate sample are typed in italics, those detected only in bile samples are in italics and underlined. Metabolites which were undetectable by NMR-based spectroscopy are indicated in gray fonts. Differences across experimental groups were investigated by one-way ANOVA, followed by Tukey’s post hoc test. *p*-values are indicated as follows: all the NMP groups vs. Native: * *p* < 0.05; ** *p* < 0.01; *** *p* < 0.001, **** *p* < 0.0001; only OxC-NMP vs. Native: § *p* < 0.05; §§ *p* < 0.01; §§§§ *p* < 0.0001. Black arrows indicate a direct relationship between two metabolites; black dotted arrows denote multiple enzyme reactions to convert one metabolite into another; red arrows link one specific metabolic pathway to a different pathway; orange dotted arrows indicate metabolite translocation from mitochondria to cytoplasm and vice versa. Adapted from KEGG reference pathways. Abbreviations: I, NADH dehydrogenase; II, NADH dehydrogenase; III, cytochrome C reductase; IV, cytochrome C oxidase; V, ATP synthase; 1-MNA, 1-methylnicotinamide; 3-HB, 3-hydroxybutyrate; 3-HPA, 3-hydroxyphenylacetate; α-KG, α-ketoglutarate; AcA, acetoacetate; ACAR, acetylcarnitine; ATP, Adenosine triphosphate; Ala, alanine; Asn, asparagine; Asp, aspartate; CoQ, coenzyme Q; Cyd, cytidine; CPT, cytidine-5′-triphosphate; CSE, cystathionine; CytC, cytochrome C; DMG, N,N-dimethylglycine; Etn, ethanolamine; FA, formic acid; FFA, free fatty acid; Fru-1,6-P2, fructose 1,6-bisphosphate; GAA, guanidinoacetate acid; GAG, glycosaminoglycans; GGC, γ-glutamylcysteine; Glc, glucose; Gln, glutammine; Glu, glutammate; Gly, glycine; GPC, glycerophosphocholine; GSH, reduced glutathione; GSSG, glutathione disulfide (oxidized form); Hcy, homocysteine; IMP, inosine 5′-monophosphate; Kyn, kynurenine; MG, methylguanidine; NMN, nicotinamide ribotide; o-P-choline, O-phosphocholine; o-P-Etn, o-phosphoethanolamine; OXPHOS, oxidative phosphorylation; PA, pantothenic acid; PC, phosphatidylcholine; PCr, phosphocreatine; PE, phosphatidylethanolamine; Ser, serine; Tau, taurine; TGL, triglycerides; THF, tetrahydrofolate; TMA, trimethylamine; TMAO, trimethylamine N-oxide; UDP-GlcNAc, UDP-N-acetylglucosamine; UDP-Gal, UDP-galactose; UTP, uridine-5′-triphosphate; UDP-Glc, uridine diphosphate-glucose; UDP-GalNAc, UDP-N-acetylgalactosamine.

**Table 1 biomedicines-10-00538-t001:** **Differentially expressed metabolites in bile samples.** Metabolites were grouped in one of the following classes: (1) increased compared to Native bile; (2) decreased compared to Native bile; (3) increased at the “start” and restored at the “end” of the NMP procedure; (4) unchanged relative to Native bile; (5) differentially modulated in the h-NMP and OxC-NMP groups. Abbreviations: 3-HB, 3-hydroxybutyrate; 3-HPA, 3-hydroxyphenylacetate; GSSG, glutathione disulfide (oxidized form); NMN, nicotinamide ribotide; TMA, trimethylamine; TMAO, trimethylamine N-oxide.

		Modulation vs. Native Bile							
		h-NMP				OxC-NMP			
Class	Metabolite	Start		End		Start		End	
**(1) Increased**									
	3-HB	↑	<0.001	↑	<0.001	↑	<0.001	↑	<0.001
	Lactate	↑	<0.010	↑	<0.010	↑	<0.001	↑	<0.050
	O-phosphocholine	=		↑	<0.050	=		↑	<0.010
	GSSG	=		↑	<0.010	=		↑	<0.050
	Glucuronic acid	=		↑	<0.010	=		↑	<0.0001
**(2) Decreased**									
	TMAO	↓	<0.001	↓	<0.010	↓	<0.001	↓	<0.001
	Creatine	↓	<0.0001	↓	<0.0001	↓	<0.0001	↓	<0.0001
	C-25 tau conjugate	↓	<0.0001	↓	<0.0001	↓	<0.0001	↓	<0.0001
	N(CH_3_)_3_ phospholipid-CHO	↓	<0.0001	↓	<0.0001	↓	<0.0001	↓	<0.0001
	Glutammate	↓	<0.0001	↓	<0.0001	↓	<0.0001	↓	<0.0001
	Hypotaurine	↓	<0.0001	↓	<0.0001	↓	<0.010	↓	<0.010
	Fumarate	=		↓	<0.010	=		↓	<0.010
	NMN	↓	<0.010	=		↓	<0.0001	↓	<0.0001
**(3) Increased at the “start” and restored at the “end”**									
	Acetate	↑	<0.050	=		↑	<0.0001	=	
	Glycerol	↑	<0.0001	=		↑	<0.050	=	
	3-HPA	↑	<0.0001	=		↑	<0.0001	=	
**(4) Unchanged**									
	Alanine	=		=		=		=	
	Hypoxantine	=		=		=		=	
	Carnitine	=		=		=		=	
**(5) Differentially modulated in the h-NMP and OxC-NMP groups**									
	DMG	↑		↑	<0.010	=		=	
	Betaine	=		↑	<0.010	=		=	
	Formate	=		=		↑	<0.001	↑	<0.050
	Taurine	=		=		↓	<0.050	↓	<0.050
	Asparagine	=		=		↑	<0.050	=	

## Data Availability

Not applicable.

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
