# Peer review of "Quantitative Metabolomics of Tissue, Perfusate, and Bile from Rat Livers Subjected to Normothermic Machine Perfusion"

_biomedicines, 2022, doi:10.3390/biomedicines10030538_

Round 1

Reviewer 1 Report

Thank you very much for allowing me to review the article entitle “NMR spectroscopy-based metabolomic profile of tissue, perfusate, and bile from rat livers subjected to normothermic machine perfusion” (biomedicines-159896). Presented for Section: Molecular and Translational Medicine, in its Special Issue “New Challenges in the Study of Liver Diseases: From Molecular Pathogenesis to Therapeutic Approaches.”

This is an interesting preclinical study investigated the metabolic outcome of the procedure analyzing rat liver tissue, bile, and perfusate samples by means of high-field (600MHz) nuclear-magnetic-resonance (NMR) spectroscopy. Using a model of normothermic machine perfusion (NMR). They used two groups, with and without oxygen carrier to the perfusion fluid. They found a broad modulation of mitochondrial metabolism during NMP that exceeds energy production and redox balance maintenance.

Comments:

In the title I recommend that you do not use acronyms.

The introduction is clear and uses adequate bibliography, the references are appropriate to the research. I recommend that the end of the introduction be the statement of the objective.

Material and methods

The methodology is appropriate to the stated objective and is clearly described.

Results:

There are structured which allows a better understanding of them. The graphs are well constructed allowing a better identification of the differences over time of the groups studied.

Discussion:

It is well thought out, reflects on the findings in comparison with the literature, and discusses the meaning of the results. I suggest that you incorporate the strengths and limitations of the research to take them into account for future studies on this topic

Author Response

  1. In the title I recommend that you do not use acronyms.

Thank you for this suggestion. We rephrased the title and excluded acronyms.

  1. The introduction is clear and uses adequate bibliography, the references are appropriate to the research. I recommend that the end of the introduction be the statement of the objective. 

We thank you for this comment. To clarify the research aim, we have added the following sentence “The present study performed a quantitative metabolomics of tissue, perfusate, and bile from rat livers subjected to ex vivo perfusion” in the Introduction section (please see page 2 of the Revised Manuscript).

  1. It is well thought out, reflects on the findings in comparison with the literature, and discusses the meaning of the results. I suggest that you incorporate the strengths and limitations of the research to take them into account for future studies on this topic.

Thank you for this suggestion. The limitations of the study have been discussed in a novel paragraph, added at the end of the “Discussion” section (please see page 11 of the Revised Manuscript).

Reviewer 2 Report

In this manuscript Lonati et al. report a detailed characterization of metabolite profiles of rat samples obtained for different machine perfusion procedures using NMR spectroscopy. The manuscript is well-written, technically sound, a piece of high quality, and is an excellent demonstration of the outstanding potential of NMR spectroscopy in metabolomics.

There are only a few minor things that should be addressed by the authors:

  • On page 5 the authors list a R2 of 0.082, is this correct?
  • NAD+: please put ‘+’ in superscript
  • In vivo/in vitro: please put in italic
  • Figure 3 and elsewhere: y-axis labeling is “Norm. concentration (mM)”; this is unclear to me, to what are the concentrations normalized?
  • Figure 5 B: please add error bars and use error propagation to calculate the SDs

Author Response

  1. On page 5 the authors list a R2 of 0.082, is this correct?

The correct value of R2 is 0.820. Thank you very much for pointing this out.

  1. NAD+: please put ‘+’ in superscript

Thank you for this advice. We have changed this in 4 positions throughout the manuscript.

  1. In vivo/in vitro: please put in italic.

The formatting of “in vivo”, “in situ”, and “ex vivo” has been changed throughout the manuscript.

  1. Figure 3 and elsewhere: y-axis labeling is “Norm. concentration (mM)”; this is unclear to me, to what are the concentrations normalized? 

We describe in the Statistics section how the quantitative metabolomics data was normalized to account for dilution effects (by using the PQN approach). This is a standard procedure in metabolomics when technically an equal tissue weight or exact volume is hard to obtain for all replicates. In order to not confuse the reader, we however excluded now the “Norm.” from all y-axis.

  1. Figure 5 B: please add error bars and use error propagation to calculate the SDs.

Thank you for this important point. The plots have been revised including the error bars and SDs.